

# Potential drug-drug interactions and their associated factors in hospitalized COVID-19 patients with comorbidities

Imanda Dyah Rahmadani[1,2], Sylvi Irawati[3,4], Yosi Irawati Wibowo[3,4] and Adji Prayitno Setiadi[3,4]

[1] Fakultas Farmasi, Universitas Surabaya, Surabaya, Indonesia
[2] Department of Pharmacy, Hospital of Muhammadiyah Lamongan, Lamongan, Indonesia
[3] Department of Clinical and Community Pharmacy, Faculty of Pharmacy, Universitas Surabaya, Surabaya, Indonesia
[4] Center for Medicines Information and Pharmaceutical Care (CMIPC; Pusat Informasi Obat dan Layanan Kefarmasian (PIOLK)), Faculty of Pharmacy, Universitas Surabaya, Surabaya, Indonesia

## ABSTRACT

**Background**. Hospitalized COVID-19 patients with comorbidities receive more complex drug therapy. This increases the probability of potential drug-drug interactions (pDDIs). Studies on pDDIs in hospitalized patients with COVID-19 in countries with limited resources like Indonesia during the later period of the disease are still limited. This study aims to identify the pattern of pDDIs in hospitalized COVID-19 patients with comorbidities and their associated factors, especially in the second wave of the disease in Indonesia.

**Methods**. This study was a longitudinal-retrospective study observing hospitalized COVID-19 patients with comorbidities using medical record data in June–August 2021 at a public hospital in a region in Indonesia. pDDIs were identified using the Lexicomp® database. Data were descriptively analyzed. Factors associated with important pDDIs were analyzed in multivariate logistic regression model.

**Results**. A total of 258 patients with a mean age of $56.99 \pm 11.94$ years met the inclusion criteria. Diabetes mellitus was the most common comorbidity experienced by 58.14% of the patients. More than 70% of the patients had one comorbidity and the average number of administered drugs was $9.55 \pm 2.71$ items per patient. Type D pDDIs, which required modification of therapeutic regimens, amounted to 21.55% of the total interactions. Only the number of drugs was significantly and independently associated with type D pDDIs (adjusted odds ratio 1.47 [1.23–1.75], $p < 0.01$).

**Conclusion**. The drugs involved in the pDDIs of hospitalized COVID-19 patients with comorbidities may differ depending on the disease periods, hospital settings, or countries. This study was small, single center, and of short duration. However, it may give a glimpse of important pDDIs during the delta variant of COVID-19 in a similar limited-resource setting. Further studies are needed to confirm the clinical significance of these pDDIs.

Corresponding author
Sylvi Irawati, syl@staff.ubaya.ac.id

## INTRODUCTION

Treatment options for the coronavirus disease (COVID-19) are evolving as new evidence emerges. In December 2020, five organizations of different medical professionals in Indonesia published a new guideline for the management of COVID-19. The selection of pharmacological management for COVID-19 is based on the level of severity of the disease *i.e.,* asymptomatic, mild, moderate, severe, and critical. The more severe the case of COVID-19, the more complex the pharmacological therapy (*Burhan et al., 2021*). One of factors associated with having more severe conditions of COVID-19 is the presence of one or more comorbidities (*Li et al., 2021*; *Bajgain et al., 2021*).

Patients hospitalized with COVID-19 are often present with one or more comorbidities (*Bajgain et al., 2021*), thus increasing the number of drugs taken and the therapy complexity. The more drugs taken by a patient, the greater the likelihood of adverse drug reaction occuring. Adverse drug reactions can result from potential drug-drug interactions (pDDIs) when the effect of one drug is modified by the presence of another (*Krska & Cox, 2019*; *Baxter, Preston & Stockley, 2013*).

Studies on pDDIs in hospitalized COVID-19 patients with or without comorbidities have been conducted in several countries including Iran, Italy, and Spain. At least one pDDIs may affect more than 60% to 90% of all hospitalized patients with COVID-19 (*Cantudo-Cuenca et al., 2021*; *Cattaneo et al., 2020*; *Larasati, Nisa & Yumna, 2021*; *Martinez-López de Castro et al., 2020*). Tools to assess pDDIs may vary across studies thus affecting how pDDIs are categorized (*Cantudo-Cuenca et al., 2021*; *Cattaneo et al., 2020*; *Larasati, Nisa & Yumna, 2021*). There are various factors that can influence the occurrence of pDDIs, including age, number of drugs, using drugs with a narrow therapeutic index, the presence of comorbidities, receiving treatment at an intensive care unit (ICU), and length of stay (*Cantudo-Cuenca et al., 2021*; *Larasati, Nisa & Yumna, 2021*; *Mahboobipour & Baniasadi, 2020*).

Research on drug interactions among hospitalized COVID-19 patients in Indonesia is still limited, including those conducted by *Larasati, Nisa & Yumna (2021)* at a hospital in West Java and by *Yuniar Ramadhiani et al. (2021)* at a hospital in Palembang, South Sumatra. Both studies were conducted during different periods of COVID-19, one was in August–December 2020 (*Larasati, Nisa & Yumna, 2021*) and the other was in April–June 2021 (*Yuniar Ramadhiani et al., 2021*). The pattern of drugs used for COVID-19 were different between the earlier and the later period of the disease following new guidelines and related-studies (*Cantudo-Cuenca et al., 2021*; *Cattaneo et al., 2020*; *Larasati, Nisa & Yumna, 2021*; *Martinez-López de Castro et al., 2020*; *Mahboobipour & Baniasadi, 2020*; *Yuniar Ramadhiani et al., 2021*). Our study aims to identify the pattern of pDDIs in hospitalized COVID-19 patients with comorbidities and the associated factors, especially in the second wave of the disease in Indonesia.

## METHODS

This was a longitudinal retrospective study conducted at a private hospital in the city of Lamongan, East Java, Indonesia. The hospital is one of the two largest hospitals in

the city where patients with COVID-19 were mainly refered to. The study was approved by the Ethics Committee of the University of Surabaya (No. 38/KE/IV/2022) and the Health Research Ethics Committee of the Hospital of Muhammadiyah Lamongan (No. 0350/KET/III.6.AU/F/2022). The Health Research Ethics Committee waived the need for consent for this study.

## Sample recruitment

We recruited patients aged $\geq$ 17 years who were admitted at the hospital from June 1st until August 31st 2021, diagnosed with COVID-19, presenting with multiple comorbidities, concomitantly prescribed with two or more medications, and hospitalized for more than 24 hours. Only patients with complete medical-record data were analysed for pDDIs.

## Data collection

Data on patients' characteristic and medications were collected from medical and pharmacy dispensing records. The types and mechanisms of pDDIs were assessed using the Lexicomp® database (UpToDate Inc., 2022). The types of pDDIs were classified according to Lexicomp® as follows: A (no interaction), B (no action needed), C (monitor therapy), D (modify regimen), and X (avoid combination). Only pDDIs categorized as type D and X were noted and considered for further analysis.

## Data analysis

Data were analyzed using IBM SPSS Statistics version 25 for Windows (IBM Corp., Armonk, N.Y., USA). Categorical and continuous data were presented as percentages and mean ($\pm$ standard deviation, SD), respectively. Univariate analysis with chi-square was used to test for any association between each patient factor (*i.e.*, gender, age, length of hospital stay, ICU admission, number of drugs, number of comorbidity, and type of comorbidity - diabetes mellitus, hypertension, cardiovascular disease, kidney disease, or other comorbidities) with pDDIs of type D/X. To explore the independent associations of patient factors with pDDIs, a binary logistic regression was performed. All the independent variables (*i.e.,* patient factors) were entered into the model. A factor with *p*-value <0.05 indicates a statistically significant association, and the odds ratio (OR) with 95% confidence intervals show a measure of association between the factor and type D/X pDDIs.

## RESULTS

There were 307 patients diagnosed with COVID-19 and comorbidities admitted at the hospital from June 1st until August 31st 2021. We included 258 patients who were $\geq$ 17 years old, concomitantly prescribed with two or more drugs, and hospitalized for more than 24 hours. The remaining patients were excluded due to a younger age ($n = 1$), a hospital stay of less than 24 hours ($n = 33$), and incomplete medical records.

### Patient characteristics

The patients' characteristics are presented in Table 1. Overall, the majority of the patients were women (53.1%) and middle aged (mean 56.99 $\pm$ 11.94 years). Most of the patients had one comorbidity (72.87%) with the most common being diabetes mellitus (58.14%).

**Table 1  Patients' characteristics on admission.**

| Patients' characteristics | Total ($N = 258$) | Men ($n = 121$) | Women ($n = 137$) | $p$-Value |
|---|---|---|---|---|
| | | Frequency, n (%) | | |
| Age (in years) | | | | |
|   17–39 | 18 (6.98) | 5 (4.13) | 13 (9.49) | 0.199 |
|   40–59 | 125 (55.43) | 58 (47.93) | 67 (48.91) | |
|   ≥60 | 115 (44.57) | 58 (47.93) | 57 (41.61) | |
| ICU admission | 76 (29.46) | 28 (23.14) | 48 (35.04) | **0.036**[*] |
| Number of comorbidities | | | | |
|   1 | 188 (72.87) | 88 (72.73) | 100 (72.99) | 0.292 |
|   2 | 63 (24.42) | 30 (24.79) | 33 (24.09) | |
|   3 | 5 (1.94) | 1 (0.83) | 4 (2.92) | |
|   4 | 2 (0.78) | 2 (1.65) | 0 | |
| Types of comorbidities | | | | |
|   Diabetes mellitus | 150 (58.14) | 62 (51.2) | 88 (64.2) | **0.035**[*] |
|   Hypertension | 62 (24.03) | 30 (24.8) | 32 (23.4) | 0.788 |
|   Cardiovascular disease | 50 (19.38) | 31 (25.6) | 19 (13.9) | **0.017**[*] |
|   Kidney disease | 42 (16.28) | 26 (21.5) | 16 (11.7) | **0.033**[*] |
|   Others[**] | 33 (12.79) | 10 (8.3) | 23 (16.8) | **0.041**[*] |
| Number of drugs | | | | |
|   2–4 | 2 (0.78) | 1 (1.65) | 1 (0.73) | 0.719 |
|   ≥5 | 256 (99.22) | 120 (99.17) | 136 (99.27) | |
| Prescribed with narrow therapeutic index drugs | 14 (5.43) | 6 (5.0) | 8 (5.8) | 0.755 |
| Length of hospital stay (in days) | | | | |
|   1–7 | 134 (51.94) | 61 | 73 | 0.645 |
|   >7 | 124 (48.06) | 60 | 64 | |
| | Mean ± SD | Median (Q1, Q3) | | |
| Number of comorbidities | 1.31 ± 0.034 | 1 (1, 2) | 1 (1, 2) | 0.973 |
| Number of drugs | 9.55 ± 2.71 | 9 (8, 11) | 9 (7.5, 11) | 0.286 |
| Length of hospital stay (in days) | 8.06 ± 4.49 | 7 (5, 11) | 7 (4, 11) | 0.329 |

**Notes.**

The $p$ values that show a statistically significant difference between groups are presented in bold.

ICU, intensive care unit; Q1, quartile 1; Q3, quartile 3; SD, standard deviation.

[**]Including chronic respiratory diseases, liver disease, disease involving immune mechanism, cancer, tuberculosis, Alzheimer, vertigo, epilepsy, or osteoarthritis.

[*]Statistically significant at $p \leq 0.05$.

Almost all patients received polypharmacy (≥5 drugs, 99.22%). The mean number of drugs received by the patients was 9.55 ± 2.71.

## Medications during hospitalization

The following drugs were received by more than 60.0% of all of the included patients during their hospital stay: methampyrone or metamizole (94.96%), N-acetylcysteine (89.92%), pantoprazole (86.82%), azithromycin (72.48%), favipiravir (67.05%), and vitamin C (60.85%). More than half of the patients were also on unfractionated heparin (59.30%) and dexamethasone (55.81%). Diabetic patients were also treated with rapid-acting insulin glulisine (25.58%), metformin (9.69%), and long-acting insulin glargine (6.98%). Antihypertensive agents used were amlodipine (18.22%), furosemide (14.73%),

and candesartan (8.14%). Aspirin and clopidogrel were used by 12.4% and 10.85% of the patients, respectively.

### Potential drug-drug interactions

In total, there were 1462 pDDIs from all of the drugs used by the patients. Of these numbers, 21.55% were type D pDDIs with at least one interaction of this type affected 73.26% (189 of 258) of the patients. The most dominant pDDIs was type C, affected 68.88%. None of the pDDIs was rated as type X. The most common mechanisms of pDDIs was a sinergistic effect of two drugs affecting coagulation system thus increased the risk of bleeding. The levels of severity type D pDDIs were major (9.21%) and moderate (90,79%). Both major and moderate type D pDDIs were mostly supported by fair documentation (75.86% for major interactions and 66.43% for moderate interactions). Only 10.34% of major interactions were supported by excellent documentation; these interactions involved dexamethasone-atracurium and aspirin-warfarin. Similarly, the reliability of documentation was excellent in 11.19% of all moderate pDDIs. Details on the drugs and mechanisms involved in the type D pDDIs are presented in Table 2.

### Factors associated with type D potential drug-drug interactions

In the univariate regression analysis, the number of drugs and length of stay were significantly associated with the odds of having type D pDDIs ($p < 0.01$). However, in the multivariate analysis, only number of drugs was found to be statistically significant (adjusted odds ratio [aOR] 1.47 (95% CI [1.23–1.75], $p < 0.01$). Details of the analyses are presented in Table 3.

## DISCUSSION

In this study, 70% of hospitalized COVID-19 patients with comorbidities were exposed to at least one type D pDDIs that required specific actions to minimize adverse effects (mean $1,67 \pm 1,18$ interaction per patient). This type of interaction represented around 20% of all types of pDDIs, with approximately 10% of those being major interactions, supported by fair-to-moderate documentations. None of the pDDIs were rated as type X. For every unit increase in number of drugs, the odds of the patients having at least one of type D pDDIs increased by a factor of 1.5. This estimation was independent of other factors and statistically significant.

The pattern of type D pDDIs in this study was different to that of a similar study involving 260 patients conducted by Mahboobipour et al. in Iran during an earlier period of COVID-19 transmission (March 2020). While we did not identify type X pDDIs, Mahboobipour et al. found type D/X pDDIs affected a smaller portion of patients in the period of March 2020 than did type D pDDIs in the period of June to August 2021 in our study (38% *vs* 70%). Type D/X pDDIs also represented a smaller portion of all types of pDDIs compared to our study (18% *vs* 20%) (*Mahboobipour & Baniasadi, 2020*). Another small study involving 107 patients conducted in a hospital in Indonesia during the period of August –December 2020 found a small portion of type X pDDIs (*Larasati, Nisa & Yumna, 2021*). These differences in pDDIs profile to our study can be caused by different drugs used during those two

**Table 2  Type D pDDIs identified in hospitalized COVID-19 patients with comorbidities.**

| Pairs of medications | Mechanisms of the pDDIs | Severity | Reliability | n (%) |
|---|---|---|---|---|
| heparin + metamizole | Enhanced anticoagulant effect of heparin | Moderate | Fair | 141 (54.65) |
| furosemide + metamizole | Diminished diuretic effect of furosemide, and enhanced nephrotoxic effect of metamizole | Moderate | Excellent | 32 (12.40) |
| aspirin + metamizole | Enhanced adverse/ toxic effect of salicylates, *i.e.*, risk of bleeding | Moderate | Good | 29 (11.24) |
| enoxaparin + metamizole | Enhanced anticoagulant effect of enoxaparin | Moderate | Fair | 19 (7.36) |
| heparin + clopidogrel | Enhanced anticoagulant effect of heparin | Moderate | Good | 14 (5.43) |
| heparin + aspirin | Enhanced anticoagulant effect of heparin | Moderate | Good | 12 (4.65) |
| enoxaparin + aspirin | Enhanced anticoagulant effect of enoxaparin | Moderate | Fair | 9 (3.49) |
| sucralfate + vitamin D3 | Increased serum concentrations of sucralfate (specifically aluminum) | Moderate | Fair | 8 (3.10) |
| heparin + sertraline | Enhanced anticoagulant effect of heparin | Moderate | Good | 5 (1.94) |
| enoxaparin + clopidogrel | Enhanced anticoagulant effect of enoxaparin | Moderate | Fair | 5 (1.94) |
| azithromycin + chlorpromazine | Enhanced the QTc-prolonging effect | Major | Fair | 4 (1.55) |
| aspirin + ticagrelor | Enhanced antiplatelet effect of ticagrelor | Major | Fair | 2 (0.78) |
| codeine + diazepam | Enhanced CNS depressant effect of codeine | Major | Fair | 2 (0.78) |
| fluoxetine + metamizole | Enhanced the antiplatelet effect of metamizole and diminished therapeutic effect of fluoxetine | Major | Good | 2 (0.78) |
| metamizole + warfarin | Enhanced anticoagulant effect of warfarin | Moderate | Fair | 2 (0.78) |
| midazolam + morphine | Enhanced CNS depressant effect of morphine | Major | Fair | 2 (0.78) |
| clopidogrel + morphine | Diminished antiplatelet effect of clopidogrel and decrease serum concentrations of clopidogrel | Major | Fair | 2 (0.78) |
| heparin + fluoxetine | Enhanced anticoagulant effect of heparin | Moderate | Good | 2 (0.78) |
| dexamethasone + atracurium | Enhanced adverse neuromuscular effect of corticosteroids, i.e., muscle weakness | Major | Excellent | 2 (0.78) |
| azithromycin + domperidone | Enhanced QTc-prolonging effect | Moderate | Fair | 1 (0.39) |
| levofloxacin + chlorpromazine | Enhanced QTc-prolonging effect | Major | Fair | 1 (0.39) |
| levofloxacin + domperidone | Enhanced QTc-prolonging effect | Moderate | Fair | 1 (0.39) |
| enoxaparin + sertraline | Enhanced anticoagulant effect of enoxaparin | Moderate | Fair | 1 (0.39) |
| enoxaparin + ticagrelor | Enhanced anticoagulant effect of enoxaparin | Moderate | Fair | 1 (0.39) |
| heparin + ketorolac | Enhanced anticoagulant effect of heparin | Moderate | Good | 1 (0.39) |
| heparin + piracetam | Enhanced anticoagulant effect of heparin | Moderate | Good | 1 (0.39) |
| acarbose + insulin glulisine | Enhanced hypoglycemic effect of insulin | Moderate | Fair | 1 (0.39) |
| acarbose + insulin glargine | Enhanced hypoglycemic effect of insulin | Moderate | Fair | 1 (0.39) |
| alprazolam + codeine | Enhanced CNS depressant effect of codeine | Major | Fair | 1 (0.39) |
| amitriptyline + tramadol | Enhanced CNS depressant effect of tramadol and serotonergic effect of amitriptyline | Major | Good | 1 (0.39) |
| aripiprazol + phenytoin | Decreased serum concentrations of aripiprazole | Major | Good | 1 (0.39) |
| aripiprazol + codeine | Enhanced CNS depressant effect of codeine | Major | Fair | 1 (0.39) |
| aspirin + warfarin | Enhanced anticoagulant effect of warfarin | Major | Excellent | 1 (0.39) |

**Table 2** (*continued*)

| Pairs of medications | Mechanisms of the pDDIs | Severity | Reliability | n (%) |
|---|---|---|---|---|
| clobazam + codeine | Enhanced CNS depressant effect of codeine | Major | Fair | 1 (0.39) |
| codeine + diphenhydramine | Enhanced CNS depressant effect of codeine | Major | Fair | 1 (0.39) |
| codeine + chlorpheniramine maleat | Enhanced CNS depressant effect of codeine | Major | Fair | 1 (0.39) |
| codeine + trifluoperazine | Enhanced CNS depressant effect of codeine | Major | Fair | 1 (0.39) |
| codeine + lorazepam | Enhanced CNS depressant effect of codeine | Major | Fair | 1 (0.39) |
| dexamethasone + phenytoin | Decreased serum concentrations of dexamethasone | Major | Fair | 1 (0.39) |
| gabapentin + tramadol | Enhanced CNS depressant effect of tramadol | Major | Fair | 1 (0.39) |

Notes.
CNS, central nervous system; n, number of patients; pDDIs, potential drug-drug interactions; QTc, corrected QT interval.

**Table 3** Factors associated with type D pDDIs.

| | Univariate analysis | | Multivariate analysis | |
|---|---|---|---|---|
| | OR (95% CI) | *p*-value | adjusted OR (95% CI) | *p*-value |
| Sex | | | | |
|     Men (*versus* women) | 1.15 (0.66–2.01) | 0.62 | 1.00 (0.53–1.91) | 0.99 |
| ICU admission | 0.76 (0.42–1.38) | 0.37 | 0.69 (0.33–1.42) | 0.31 |
| Types of comorbidities | | | | |
|     Diabetes mellitus | 0.878 (0.50–1.55) | 0.65 | 0.10 (0.34–2.92) | 0.99 |
|     Hypertension | 1.68 (0.83–3.39) | 0.15 | 1.31 (0.41–4.21) | 0.66 |
|     Cardiovascular disease | 0.91 (0.46–1.81) | 0.78 | 0.51 (0.15–1.76) | 0.29 |
|     Kidney disease | 1.34 (0.60–2.97) | 0.48 | 2.19 (0.63–7.63) | 0.22 |
| Age | 1.01 (0.99–1.04) | 0.27 | 1.02 (0.99–1.05) | 0.20 |
| Number of comorbidities | 1.30 (0.75–2.25) | 0.35 | 1.07 (0.35–3.29) | 0.91 |
| Number of drugs | **1.50 (1.29–1.74)** | **0.00**[*] | **1.47 (1.23–1.75)** | **0.00**[*] |
| Length of hospital stay | **1.19 (1.09–1.29)** | **0.00**[*] | 1.07 (0.96–1.18) | 0.22 |
| Prescribed with narrow therapeutic index drugs | 4.95 (0.64–38.57) | 0.13 | 5.05 (0.58–43.94) | 0.14 |

Notes.
The estimates that are statistically significant are presented in bold.
ICU, intensive care unit
[*]Statistically significant at $p \leq 0.05$.

periods. In the early days of COVID-19 transmission, drugs that have higher risk of type X pDDIs such as chloroquine, hydroxychloroquine, remdesivir, ritonavir/lopinavir, or tocilizumab were used (*Larasati, Nisa & Yumna, 2021*; *Martinez-López-de Castro et al., 2020*; *Mahboobipour & Baniasadi, 2020*). During the period of July–August 2021, our hospital no longer used chloroquine or hydroxychloroquine as a COVID-19 therapy. Other antivirals such as lopinavir/ritonavir had never been prescribed for COVID-19 cases at our hospital.

The proportion of women involved in this study is greater than men. Several studies have shown that the majority of COVID-19 patients are men (*Cantudo-Cuenca et al., 2021*; *Larasati, Nisa & Yumna, 2021*; *Mahboobipour & Baniasadi, 2020*). While women have higher macrophage and neutrophil activity than men, the concentration of angiotensin converting enzyme 2 (ACE 2) receptor, which is the entry point for COVID-19 causing virus, is higher in male kidneys than in females; hence, men have a greater risk for

COVID-19 transmission (*Kopel et al., 2020*). In the study conducted by Gunadi et al. in COVID-19 patients, infections with the delta variant were dominated by women, whereas infections with the non-delta variant were dominated by men. However, there was no significant relationship between sex and risk of COVID-19 infection (*Gunadi et al., 2021*). The higher percentage of women than men in our study could also have been influenced by the comorbidities, of which diabetes mellitus was dominant. Women have a higher risk of developing diabetes mellitus than men, which is mostly due to hormonal biological factors related to the metabolic system (*Eling et al., 2018*). In our study, more women presented with diabetes mellitus compared to men, whereas, more men presented with cardiovascular disease and kidney disease than women. However, both number of comorbidities and number of drugs did not statistically different between men and women. In the multivariate analysis, only the number of drugs was significantly associated with type D pDDIs in our study.

Around 50% of the patients involved in our study had to stay in hospital for 1 to 7 days. The time span of data collection in this study coincided with the occurrence of the second wave of COVID-19 in Indonesia, during when the delta variant was dominant. In another study in Indonesia during August–December 2020, more than 95% of patients had a length of stay of 6 days (*Larasati, Nisa & Yumna, 2021*). One of several factors significantly associated with longer hospitalization was disease severity (*Wang et al., 2022*). In our study, however, the severity of COVID-19 was not recorded in the medical records. Another approach that has been taken to estimate the disease severity was by counting the number of patients who needed intensive treatment in the ICU. As many as 29.46% of all patients observed in this study were admitted to the ICU during hospitalization. Treatment in the ICU can be an indicator of the severity of COVID-19, but it is also necessary to consider that the data collected in our study were from the period of July–August 2021, which corresponded to the second wave of COVID-19 in Indonesia, when there was a surge of the number of patients admitted to the hospital, while the ICU capacity was limited.

Diabetes mellitus was the most common comorbidity found in the patients included in this study. Patients with diabetes mellitus have decreased phagocytic cell function, high levels of the proprotein convertase furin, and increased expressions of ACE-2 receptor. Furin facilitates the activation of the SARS-CoV-2 S (spike) protein that attaches to the ACE-2 receptor, thereby allowing the virus to enter the host cell. In patients with diabetes mellitus, there are also impaired T-cell function and increased concentrations of interleukin-6. The second most common comorbidity was hypertension. In hypertensive patients, there is upregulated formation of ACE-2 (*Ejaz et al., 2020*). The third most common type of comorbidity was cardiovascular disease. Cardiovascular disease, in this study, did not include hypertension, but including any of the following: ischemic heart disease, cerebrovascular accident, or other diagnosed cardiovascular-related disease. The types of comorbidities found in our study are in agreement with previous studies with though slightly different rankings and also not including obesity (*Larasati, Nisa & Yumna, 2021*; *Djaharuddin et al., 2021*). The relationship between cardiovascular disease and COVID-19 is not known with certainty, but is thought to involve high expressions of ACE-2 receptors on heart muscle cells and decreased immune system (*Ejaz et al., 2020*).

Our study did not find an association between diabetes mellitus and type D pDDIs. This is in line with the similar study from Iran. Despite a significant association between diabetes mellitus and pDDIs in univariate analysis, there was no significant association between those variables in multivariate analysis (*Mahboobipour & Baniasadi, 2020*). The lack of significant association might be caused by the lack of interactions between antidiabetic medicines and medicines for COVID-19.

The types of comorbidities most commonly found in this study support the pattern of pDDIs identified in the patients. In this study, drugs affecting coagulation system, which are often used in treating cardiovascular disease, were also the most common drugs involved in type D pDDIs. More than 50% of pDDIs observed in this study involved a combination of heparin and metamizole which was categorized as a moderate interaction with a fair reliability of documentation. This type of interaction would increase the risk of bleeding in patients. However, due to the nature of our data collection, we were not able to ascertain whether the interaction resulted in actual bleeding. There are limited studies on metamizole, including heparin-metamizole interaction. Although metamizole is still largely used in Asian countries, including Indonesia, this medicine has been long withdrawn from the market in the United States and other European countries (*Andrade et al., 2016*). *In vivo*, 1 g of metamizole given intravenously had an antiplatelet effect *via* blockade of type TXA2 cyclooxygenase-1. However, the effect was short-lasting (*Papp et al., 2014*). While there is a need to monitor the effect of this type of pDDIs, more studies on the clinical significance of this particular interaction are needed.

Similar to previous studies, the patients in our study received more than five drugs per patient on average. It has been known that multidrug treatment is one of the risk factors of DDIs and adverse drug reactions (*Krska & Cox, 2019*; *Baxter, Preston & Stockley, 2013*). However, the magnitude of the risk might be different in different subgroups of population, including in patients with COVID-19. In our study, the number of drugs was independently and significantly associated with type D pDDIs. The results showed that adding one item of drug was associated with a 1.5 times higher risk of type D pDDIs. This result is in line with previous study involving 205 COVID-19 patients during the period of March 1st to April 30th 2020 in a tertiary hospital in Spain. The study showed that the number of drugs prescribed was associated with real DDIs (adjusted OR 1.42, 95% CI [1.12–1.81]) (*Cantudo-Cuenca et al., 2021*). The small study from Iran showed that ischemic heart disease and ICU admission were also factors associated with type D pDDIs. However, this study did not include the number of drugs in its analyses (*Mahboobipour & Baniasadi, 2020*). The small study conducted in a hospital in Indonesia also showed a significant correlation between two variables, *i.e.,* number of drugs and comorbidities, and the occurrence of all types of pDDIs (*Larasati, Nisa & Yumna, 2021*). Based on studies we traced for comparison, it seems that studies on pDDIs in COVID-19 patients in larger setting are still limited.

The inclusion of only type D/X pDDIs in our study was based on the higher risk of these types of interactions to the patients. Type D/X pDDIs need to be intervened or avoided that it also requires more consideration from healthcare professionals. By identifying these types of pDDIs and their associated factors we would be able to develop a strategy to

prevent pDDIs which pose the most risk for the patients. Although the strategy may not encompass all spectrum of risk, this prioritization is important in a setting where resources are limited.

In this study, the use of Lexicomp®to assess pDDIs was based on several consideration such as availability, easy-to-use, up-to-date evidence for interactions, frequently used in studies on DDIs, and supporting research. Two studies concluded that Lexicomp® and Micromedex showed the best performance to analyze pDDIs (*Patel & Beckett, 2016*; *Kheshti, Aalipour & Namazi, 2016*). However, a certain medicine such as metamizole, which is commonly used in Indonesia, is only available in Lexicomp®database. Despite these benefits, there are two complementary medicines that are not listed in the Lexicomp® database, namely *Curcuma*, which contains *Curcuma xanthorrhiza* extract, and plasmin, which contains *Lumbricus rubellus* extract. However, these drugs were little prescribed to the patients.

There are several limitations of our study. Factor such as obesity could not be observed due to its unavailability in patients' medical records. Despite obesity is common in patients with COVID-19, its association to pDDIs has not been found significant (*Mahboobipour & Baniasadi, 2020*). Our study was based on a small number of patients with short duration of recruitment and in a single clinical setting. However, the study might give an impression on important types of pDDIs occurred in a hospital in a developing country like Indonesia where resources, including medicines, were limited especially during the delta variant of COVID-19. Another limitation is we were not able to follow the clinical consequence of type D pDDIs due to difficulties in conducting the research prospectively and in using an adverse-effect reporting system during the period of COVID-19 with delta variant. During this period, the hospital was overwhelmed that it became challenging to identify, prevent, record, and manage important types of pDDIs.

## CONCLUSIONS

The drugs involved in pDDIs in hospitalized COVID-19 patients with comorbidities may differ between earlier and later periods of the disease, and when comparing different hospital settings, or countries. Safer drugs seem to be used in the later period of the disease, thus avoiding the riskier type of pDDIs, although the variant of SARS-CoV-2 is more infectious or harmful. The number of drugs is still the main factor independently associated with an important type of pDDIs. This study is small, single center, and of short period thus the results might be different than in large multicenter settings. However, the study gives a glimpse of type D pDDIs observed during the period of delta variant of COVID-19 in a similar limited-resource setting. The clinical significance of these pDDIs needs further studies.

## ACKNOWLEDGEMENTS

The authors would like to thank Muhammadiyah Lamongan Hospital for all supports while conducting the data collection.

### Funding

This work was supported by the Ministry of Education, Culture, Research, and Technology of the Republic of Indonesia (No. 048/SP-Lit/LPPM-01/KemendikbudRistek/Mono/FF/V/2022). The funders had no role in study design, data collection and analysis, decision to publish, or preparation of the manuscript.

### Grant Disclosures

The following grant information was disclosed by the authors:
The Ministry of Education, Culture, Research, and Technology of the Republic of Indonesia: 048/SP-Lit/LPPM-01/KemendikbudRistek/Mono/FF/V/2022.

### Competing Interests

The authors declare there are no competing interests.

### Author Contributions

- Imanda Dyah Rahmadani conceived and designed the experiments, performed the experiments, analyzed the data, prepared figures and/or tables, authored or reviewed drafts of the article, and approved the final draft.
- Sylvi Irawati conceived and designed the experiments, analyzed the data, prepared figures and/or tables, authored or reviewed drafts of the article, and approved the final draft.
- Yosi Irawati Wibowo conceived and designed the experiments, authored or reviewed drafts of the article, and approved the final draft.
- Adji Prayitno Setiadi conceived and designed the experiments, authored or reviewed drafts of the article, and approved the final draft.

### Human Ethics

The following information was supplied relating to ethical approvals (i.e., approving body and any reference numbers):

The Health Research Ethics Committee of the Hospital of Muhammadiyah Lamongan and the Ethics Committee of the University of Surabaya granted Ethical approval to carry out the study ((No. 0350/KET/III.6.AU/F/2022 and No. 38/KE/IV/2022, respectively).

### Ethics

The following information was supplied relating to ethical approvals (i.e., approving body and any reference numbers):

The Health Research Ethics Committee of the Hospital of Muhammadiyah Lamongan and the Ethics Committee of the University of Surabaya granted Ethical approval to carry out the study ((No. 0350/KET/III.6.AU/F/2022 and No. 38/KE/IV/2022, respectively).

### Data Availability

The raw data is available in the Supplemental File.

## Supplemental Information

Supplemental information for this article can be found online at http://dx.doi.org/10.7717/peerj.15072#supplemental-information.

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
