# Peer review of "Potential drug-drug interactions and their associated factors in hospitalized COVID-19 patients with comorbidities"

_PeerJ, doi:10.7717/peerj.15072_

## Round 0.1 · original submission · Major Revisions

Dear authors, thank you for your submission. Your manuscript has been accepted conditionally upon major revisions.

Please, revise and address the reviewers' comments. Make sure to clarify methods and methodological decisions as well as properly discuss your results.

Reviewer 1 ·

Basic reporting

No Comment

Experimental design

No Comment

Validity of the findings

No Comment

Additional comments

• Methodology: authors stated “This study is a cross-sectional study observing hospitalized COVID-19 patients with comorbidities using medical records”- whether data collection was Cross-sectional prospective or retrospective?
• But from the Discussion it is understood that the data collection was retrospective (“In our study, we were not able to observe some variables due to their unavailability in the 230 patients'’ medical records or the Lexicomp® database” Lines 229 &230).
• Since more than 50% of pDDIs are related to an increased risk of bleeding- have you noticed any such bleeding reports?
• It would have added more value to this study, if authors follow up with the patients and recorded adverse events with D type DDIs.

Reviewer 2 ·

Basic reporting

1. Overall the structure of the manuscript organisation is ok but the study conclusion is based on a small number of subjects, in a single clinical setting, and the duration of recruitment was only 3 months. These limitations should be brought up in the abstract and conclusions.
2. In a larger clinical setting or in a public hospital setting, the outcomes could be different that we do not know. The authors should justify their outcomes broadly in their discussion section in compare with similar studies in a different context and may be in a different country.

Experimental design

1. The age range of the subjects should be specific NOT just >17 years, it should be a range.
2. Why the authors chose type-D and X pDDIs (according to Lexicomp) were considered. Why they did not anayzed data of other other interactions? Please discuss.
3. Why they chose Lexicomp scale? Please discuss

Validity of the findings

1. Diabetes is a common known comorbidity of SARS-CoV-2 infection and authors are correct. However, authors could connect diabetes, obesity properly with pDDIs. They should include some supporting data from the literature.
2. Multidrug treatments are associated with pDDIs. Authors could explain their findings broadly in this context.

Additional comments

The limitations should be broadly written in the conclusion section.

---

## Round 0.2 · Minor Revisions

Dear authors, thank you for your revised version. Before acceptance, I noticed in table 1 that you considered all the statistics related to the total sample (n=258), however the majority of your data sample belongs to females. It would be interesting and useful (since it was quite discussed in your manuscript and is a difference to other works) to include in the table the values for the normalisation to the specific male/female data sample, ie, how many males had diabetes, hypertension, etc... and how many females had diabetes, hypertension, were over 60 yrs old, etc etc . Otherwise, please state in your manuscript why the gender differences have no been considered in your analysis (including univariate, multivariate regression; tables 2, 3).

Also, please consider the attachments sent to you.

Otherwise, great work, and I looking forward to seeing your revision.

Reviewer 1 ·

Basic reporting

No Comment

Experimental design

No Comment

Validity of the findings

No Comment

Additional comments

NIL

Reviewer 2 ·

Basic reporting

Overall the manuscript has improved. The authors addressed my comments.

Experimental design

The explanation and modification in the manuscript are reasonable.

Validity of the findings

Reasonable.

Additional comments

Authors have addressed my comments in the revised manuscript.

---

## Round 0.3 · accepted · Accept

Dear authors, once again thank you for your submission and congratulations to get it approved for publication. I hope that you consider PeerJ in the future. All the best, Sonia